# Intrinsically Adaptive Neuronal Dynamics
# Drive Accurate and Efficient Spiking Neural Networks

## Abstract

Neuronal dynamics are fundamental to the temporal processing capabilities of spiking neural networks (SNNs), directly influencing their responsiveness and integration of time-varying input features. While most existing advances in SNNs focus on architectural optimizations or learning algorithms, the potential of adaptive neuronal dynamics remains underexplored. Inspired by intrinsic regulation mechanisms in biological neurons, we propose a novel neuron model that unifies a learnable base of neuronal timescale with intrinsically modulated dynamic regulation, enabling adaptive temporal dynamics that combine long-term stability and short-term adaptability within a computationally efficient formulation. Experiments on multiple benchmarks, including physiological signals (DEAP) and neuromorphic vision datasets (DVS-Gesture, CIFAR10-DVS), demonstrate superior performance in classification accuracy, efficiency and robustness. In particular, our model achieves state-of-the-art accuracy of 90.11% and 92.57% on the DEAP valence and arousal classification tasks, representing improvements of +9.08% and +9.97% over the baseline. These results establish adaptive neuronal dynamics as a new pathway for developing effective and efficient neuromorphic systems.

## 1. Introduction

Spiking neural networks (SNNs) have attracted growing attention due to their event-driven operation and temporal coding capabilities (Li et al., 2021a; Zhou et al., 2021), which naturally support energy-efficient computation and the processing of dynamic, time-varying signals (Maass, 1997; Kasabov et al., 2013; Yu et al., 2020; Sun et al.,

2022). A common observation in biological spiking systems is that neural computation typically operates under sparse and low-rate spiking activity, a regime closely associated with metabolic efficiency and stable information processing (Attwell & Laughlin, 2001; Lennie, 2003; Laughlin, 2001). These properties make SNNs particularly appealing for applications requiring rapid response and low power consumption, especially when integrated with neuromorphic hardware (Farsa et al., 2019; Wu et al., 2022).

Neuronal dynamics, describing the time-varying electrical behavior and state evolution of spiking neurons, form the computational foundation of spiking neural networks and play an essential role in determining their information processing capabilities. These dynamics are profoundly influenced by intrinsic cellular properties, among which the membrane time constant $\tau$ plays a particularly pivotal role (Abbott & Dayan, 2001; Gerstner & Kistler, 2002; Izhikevich, 2007). The choice of $\tau$ therefore critically influences the temporal dynamics, responsiveness, and information processing capabilities of SNNs.

Despite this importance, a common practice in most SNNs is to treat the membrane time constant as a fixed hyperparameter or, in limited cases, a globally learnable scalar (Wu et al., 2018; Fang et al., 2021; Perez-Nieves et al., 2021). This static setup significantly constrains the adaptability of neurons, limiting their ability to dynamically modulate membrane potential leakage in response to varying temporal patterns, input strengths, or spiking history. As a result, this limits their ability to process a wide range of temporal data effectively. In contrast, biological neurons exhibit dynamic neuronal timescales that continuously adapt to their input conditions and firing activities (Koch et al., 1996). Experimental studies have shown that strong synaptic drive or high firing rates can accelerate membrane decay (Kuhn et al., 2004; Ha & Cheong, 2017), while weak inputs or sparse firing result in longer integration times, thereby ensuring information processing across multiple timescales (Gerstner & Kistler, 2002). This biological evidence suggests that dynamic adjustment of the neuronal timescales is a crucial mechanism for temporal adaptability.

Early dynamic neuronal timescales mechanisms can be traced back to biophysical neuron models, such as

---

[1]Anonymous Institution, Anonymous City, Anonymous Region, Anonymous Country. Correspondence to: Anonymous Author <anon.email@domain.com>.

Preliminary work. Under review by the International Conference on Machine Learning (ICML). Do not distribute.

Hodgkin–Huxley formulations and conductance-based neuron model, where input-dependent conductance changes naturally yield adaptive membrane neuronal timescales (Hodgkin & Huxley, 1952; Destexhe, 1997; Gerstner & Kistler, 2002; Gütig & Sompolinsky, 2009). More recently, Liquid Time-Constant (LTC) networks dynamically compute time constant via parameterized recurrent functions, enabling flexible temporal integration (Hasani et al., 2021; Yin et al., 2023). Following this direction, the Brain-inspired Adaptive LIF (BA-LIF) model introduces input-driven modulation of the time constant in spiking neurons (Zhang et al., 2025a). Despite their biological plausibility or functional adaptability, these approaches share a common limitation: the dynamic mechanism introduce substantial computational overhead, which increase energy cost and hinder deployment in large-scale neuromorphic systems. Moreover, these methods primarily focus on input-driven instantaneous adaptation, overlooking the potential of learnable long-term plasticity in neuronal timescales that could be optimized globally for specific tasks. This limitation motivates our design of a lightweight, dynamic and learnable temporal mechanism that achieves high adaptability while maintaining low computational overhead.

To overcome the above limitations, inspired by biological neurons that exhibit both long-term plasticity (Rimmer & Harper, 2006; Hong et al., 2016) and short-term dynamic modulation of their membrane neuronal timescales (Kuhn et al., 2004; Ha & Cheong, 2017), we propose a novel Leaky Integrate-and-Fire neuron model with Adaptive Neuronal Dynamics, named AND-LIF, that enables both dynamic modulation and learned adaptation of the neuron's temporal characteristic. Each neuron is equipped with a trainable base neuronal timescale that captures long-term plasticity, with its value dynamically modulated based on recent inputs and spiking activity. This design enables adaptive neuronal dynamics by allowing neurons to continuously adjust their temporal integration behavior in response to incoming signals and firing history, thereby achieving short-term adaptability while maintaining long-term stability.

Since our proposed AND-LIF neuron is designed to enhance the temporal adaptability of spiking neurons via dynamic $\tau$, it is particularly suited for challenging tasks containing rich temporal dynamics. Therefore, in our experiments we focus on dynamic benchmarks, including electroencephalography (EEG) dataset DEAP (Koelstra et al., 2011) as well as the neuromorphic benchmarks such as DVS-Gesture (Amir et al., 2017) and CIFAR10-DVS (Li et al., 2017). Across these datasets, our method consistently improves classification accuracy and maintains low computational overhead, demonstrating its efficacy in enhancing the overall performance of SNNs with biologically inspired short-term adaptation and long-term stability. The main contributions of this work are threefold:

- We propose a biologically inspired spiking neuron with dynamic neuronal timescales, in which temporal integration is jointly modulated by external inputs and the neuron's own spiking activity.

- We introduce a unified design that combines a learnable intrinsic neuronal timescale with dynamic, activity-dependent modulation, allowing neurons to preserve long-term stability through learning while exhibiting short-term adaptive temporal responses.

- Extensive experiments demonstrate that the proposed AND-LIF neuron achieves state-of-the-art accuracy of 90.11% and 92.57% on the three-class valence and arousal classification tasks of the DEAP dataset, and consistently improves performance on DVS-Gesture and CIFAR10-DVS, while maintaining high energy efficiency.

## 2. Related Work

### 2.1. Static Parametrization of Neuronal Dynamics

The LIF model serves as the computational cornerstone of most contemporary spiking neural networks, balancing simplicity with reasonable biological fidelity (Maass, 1997; Yu et al., 2020; Rajakumari & Pradhan, 2022; Mishra et al., 2025; Deng et al., 2025). The introduction of surrogate gradient methods resolved the fundamental issue of non-differentiable spike generation, enabling gradient-based training and paving the way for optimizing intrinsic neuronal parameters (Neftci et al., 2019; Wu et al., 2018; Zenke & Ganguli, 2018). This led to the development of parametrized variants such as the Parametric LIF (PLIF) neuron, where the membrane time constant is learned during training (Fang et al., 2021). Extensions including layer-wise and spatio-temporally mapped constants further refined this approach (Rathi & Roy, 2021; Zhang et al., 2025b). Nonetheless, these models share a core limitation: the neuronal timescale remains static during inference. Once training concludes, the neuron's temporal dynamics can no longer adapt to real-time variations in input streams, restricting their ability to process non-stationary sensory signals.

### 2.2. Spike-Dependent Regulation of Neuronal Dynamics

Many SNN neuron adaptations implement intrinsic negative feedback through activity-dependent state variables. A representative output-driven line is spike-history adaptation via adaptive thresholds, as used in ALIF (Bellec et al., 2018) neurons and learning frameworks such as e-prop (Bellec et al., 2020), where spikes raise an internal adaptation state that suppresses subsequent firing. Hardware-oriented variants such as DEXAT (Shaban et al., 2021) follow a similar principle by introducing explicit threshold dynamics.

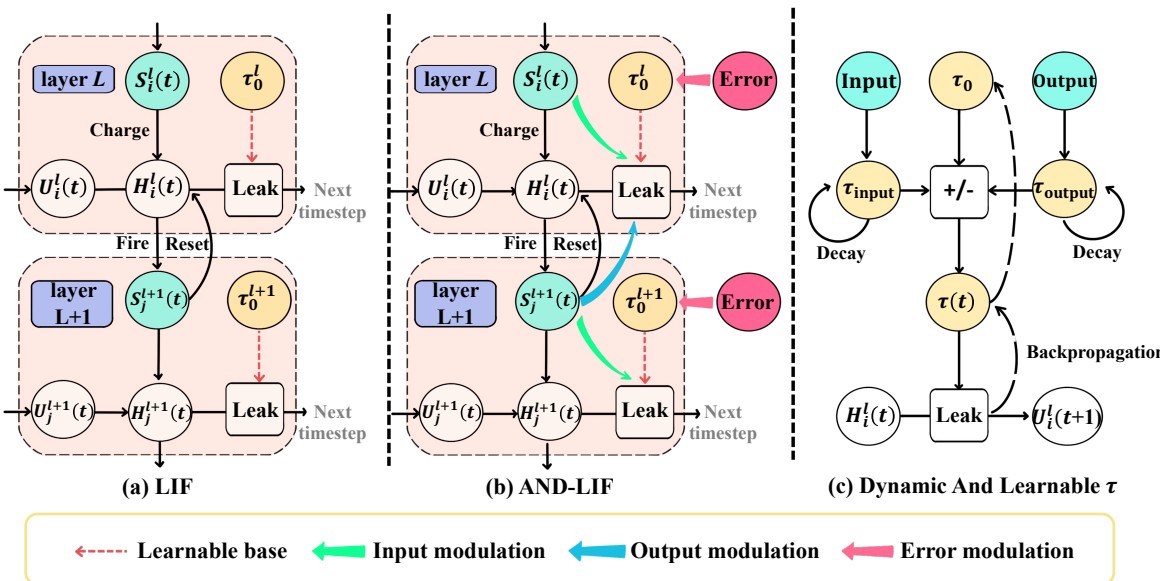

*Figure 1.* Comparison of (a) Leaky Integrate-and-Fire (LIF) neuron and (b) our proposed LIF with Adaptive Neuronal Dynamics (AND-LIF) neuron. Unlike LIF, which uses a fixed neuronal timescale, the neuronal timescale of AND-LIF is adaptively modulated by both input and output spike activities. (c) Illustration of adaptive membrane time constant $\tau(t)$, which is jointly determined by a learnable base constant $\tau_0$, input-driven modulation $\tau_{\text{input}}$, and spike-driven feedback $\tau_{\text{output}}$.

Complementarily, input-history-dependent regulation is often modeled at the synapse level via short-term synaptic plasticity (Tsodyks et al., 1998; Yu et al., 2025), which modulates synaptic efficacy under sustained drive. In contrast, our method applies input- and output-driven feedback directly to the membrane neuronal timescale, thereby regulating temporal integration rather than threshold or synaptic transmission, and can be computed with lightweight local recursions.

### 2.3. Biophysical Foundations of Adaptive Dynamics

The mechanism of adaptive membrane neuronal timescales originates in classical biophysical neuron models. In formulations such as the Hodgkin–Huxley model (Hodgkin & Huxley, 1952) and conductance-based models (Destexhe, 1997; Bai et al., 2025), input-dependent changes in synaptic or ion channel conductance naturally produce dynamic temporal integration (Gerstner & Kistler, 2002). While these models provide a mechanistic, interpretable foundation for neuronal adaptation, they require the numerical solution of complex differential equations, rendering them computationally prohibitive for large-scale network simulation and efficient neuromorphic implementation.

### 2.4. Dynamic Modulation via Structural Intervention

To enable online adaptive behavior, recent studies introduce auxiliary computations or architectural mechanisms to generate momentary time constants conditioned on inputs or internal states. Models such as Liquid Time-Constant Networks (Hasani et al., 2021; Yin et al., 2023) and related bio-inspired variants (Zhang et al., 2025a) compute dynamic time constants using additional neural layers that process inputs. While achieving short-term input-dependent adaptation, these methods introduce substantial structural overhead and computational complexity. Moreover, most such designs primarily emphasize input-driven instantaneous modulation, leaving output-driven self-regulation based on spiking history comparatively underexplored.

## 3. Method

This section introduces our proposed neuron model. We start from the commonly used leaky integrate-and-fire (LIF) neuron and explain its limitations in static neuronal dynamics. To address these limitations, we take inspiration from conductance-based spiking neuron models and propose our AND-LIF, which achieves adaptive neuronal dynamics with high computational efficiency.

### 3.1. LIF Neuron with Constant Neuronal Timescales

The LIF neuron model is commonly used due to its high computational efficiency. However, its temporal dynamics are governed by a fixed membrane neuronal timescale, which limits its ability to capture adaptive neuronal behaviors. As illustrated in Fig. 1(a), the LIF neuron integrates incoming synaptic current $I^l(t)$ into its membrane potential $U^l(t)$ while exhibiting a gradual leakage over time. The discrete-time neuronal dynamics can be formulated as follows:

$$I^l(t) = f\left(W^l, S^{l-1}(t)\right), \tag{1}$$

$$U^l(t) = \rho_{\mathrm{m}}\left(H^l(t-1) - S^l(t-1)V_{\mathrm{th}}\right), \tag{2}$$

$$H^l(t) = U^l(t) + I^l(t), \tag{3}$$

$$S^l(t) = \Theta\left(H^l(t) - V_{\mathrm{th}}\right) = \begin{cases} 1, & H^l(t) \geq V_{\mathrm{th}} \\ 0, & H^l(t) < V_{\mathrm{th}} \end{cases}, \tag{4}$$

where $S^l(t)$ and $H^l(t)$ represent the spike sequence and the membrane potential at time step $t$ for the $l$-th layer, respectively. $V_{\mathrm{th}}$ is the firing threshold that determines whether $H^l(t)$ triggers a spike, and $f(\cdot)$ denotes the convolutional or fully connected operation. $W^l$ is the synaptic weight matrix between adjacent layers. The parameter $\rho_{\mathrm{m}}$ is the membrane decay factor that controls the retention of past membrane potential at each timestep. The membrane decay factor $\rho_{\mathrm{m}}$ is intrinsically linked to the membrane time constant $\tau$ through an exponential decay relationship:

$$\rho_{\mathrm{m}} = e^{-\frac{\Delta t}{\tau}}, \tag{5}$$

where $\Delta t$ denotes the simulation timestep.

Notably, the membrane decay factor $\rho_{\mathrm{m}}$ remains fixed across time steps, which restricts the LIF neuron to a single, static temporal integration timescale. As a result, the neuron lacks the flexibility to adapt its dynamics in response to varying input intensity and spiking activity, limiting its ability to capture diverse temporal patterns in dynamic signals.

### 3.2. Conductance-Based Neuron with Adaptive Neuronal Timescales

The LIF neuron is a current-based neuron model (Maass, 1997; Yu et al., 2025), in which synaptic inputs are injected as additive currents and the neuronal timescale is fixed. In contrast, neuron models with higher biological fidelity are conductance-based (Hodgkin & Huxley, 1952; Gütig & Sompolinsky, 2009; Bai et al., 2025), and they explicitly model synaptic inputs as modulations of membrane conductance. Such conductance-based neurons capture state-dependent membrane dynamics and yield variable effective membrane integration timescales. A representative conductance-based formulation is given as:

$$C_m \frac{dU(t)}{dt} = -g_L\left(U(t) - E_L\right) - g_{\mathrm{syn}}(t)\left(U(t) - E_{\mathrm{syn}}\right) + I(t), \tag{6}$$

where $C_m$ denotes the membrane capacitance, $g_L$ the leak conductance, and $g_{\mathrm{syn}}(t)$ the total synaptic conductance. By identifying the intrinsic membrane time constant as $\tau_0 = C_m/g_L$, the effective membrane time constant $\tau_{\mathrm{eff}}$ can be written as (Gütig & Sompolinsky, 2009; Bai et al., 2025),

$$\tau_{\mathrm{eff}}(t) = \frac{C_m}{g_L + g_{\mathrm{syn}}(t)} = \frac{1}{\frac{1}{\tau_0} + \frac{g_{\mathrm{syn}}(t)}{C_m}}. \tag{7}$$

In conductance-based neuron models, synaptic conductances are dynamically modulated by both external inputs and neuronal activity (Kuhn et al., 2004; Rudolph et al., 2004). This process can be expressed as a first-order dynamical system (Górski et al., 2021):

$$\tau_{\mathrm{syn}} \frac{dg_{\mathrm{syn}}(t)}{dt} = -g_{\mathrm{syn}}(t) + \Phi\left(I(t), S(t)\right), \tag{8}$$

where $\Phi(\cdot)$ characterizes how input signals and neuronal activity jointly influence the effective synaptic conductance. The solution of Eq. (8) admits the convolution form

$$g_{\mathrm{syn}}(t) = \frac{1}{\tau_{\mathrm{syn}}} \int_{-\infty}^{t} e^{-(t-s)/\tau_{\mathrm{syn}}} \Phi\left(I(s), S(s)\right) ds$$
$$\triangleq \mathcal{G}\left(I(t), S(t)\right), \tag{9}$$

where $\mathcal{G}\left(I(t), S(t)\right)$ denotes an input- and output-dependent mapping induced by synaptic conductance dynamics. After substituting Eq. (9) into Eq. (7), we obtain the following expression for the effective membrane time constant:

$$\tau_{\mathrm{eff}}(t) = \frac{1}{\frac{1}{\tau_0} + \frac{1}{C_m} \mathcal{G}\left(I(t), S(t)\right)}, \tag{10}$$

Eq. (10) makes explicit that the effective membrane time constant depends on synaptic input and neuronal activity. However, in conductance-based neuron models, this dependence is realized indirectly through the interaction of multiple coupled dynamical states, requiring the explicit tracking of synaptic conductances and the solution of continuous-time dynamics. As a result, such models incur higher computational overhead and are less suitable for large-scale spiking neural networks and hardware-oriented implementations, thus motivating the design of a new model as we proposed in the follow.

### 3.3. AND-LIF Neuron with Adaptive Neuronal Timescales

Conductance-based neuron models exhibit a state-dependent effective membrane time constant, motivating a formulation

*Table 1.* Performance comparison on the DEAP dataset. Valence and Arousal refer to the two primary affective dimensions in emotion recognition. The experiments are conducted on both 2-class and 3-class classification settings.

| Dataset | Method | Class | Valence (%) | Arousal (%) |
|---|---|---|---|---|
| DEAP | Transfer learning (Yan et al., 2022) | 2 | 82.75 | 84.22 |
| | SNN+IIR (Xu et al., 2024) | 2 | 61.15 | 53.86 |
| | EEGNet (Gong et al., 2023) | 2 | 81.14 | 76.82 |
| | SGLNet (Gong et al., 2023) | 2 | 90.01 | 89.41 |
| | SCNN (Islam et al., 2021) | 3 | 70.23 | 70.25 |
| | DH-SNN (Zheng et al., 2024) | 3 | 77.46 | 80.23 |
| | BSNN (Sun et al., 2025) | 3 | 81.03 | 82.60 |
| | **AND-LIF (ours)** | 2 | **92.35** | **94.31** |
| | | 3 | **90.11** | **92.57** |

*Table 2.* Performance comparison on neuromorphic datasets. * indicates results from our implementations using their released code. † denotes baseline methods that use a temporal attention module to enhance performance.

| Dataset | Method | Architecture | T | Accuracy (%) |
|---|---|---|---|---|
| DVS-Gesture | STBP (Wu et al., 2018) | 5Conv,3FC | 10 | 87.50* |
| | BA-LIF (Zhang et al., 2025a) | 5Conv,1FC | 20 | 97.90† |
| | SLTT (Meng et al., 2023) | VGG-11 | 5 | 92.02* |
| | SSNN (Ding et al., 2024b) | VGG-9 | 5 | 90.74 |
| | TRR (Zuo et al., 2024) | VGG-9 | 5 | 91.67 |
| | Spikformer (Zhou et al., 2022) | Spikingformer | 4 | 93.40* |
| | **AND-LIF (ours)** | 5conv,3FC | 10 | 96.18 |
| | | VGG-11 | 5 | 95.14 |
| | | Spikingformer | 4 | 94.79 |
| | | Spikingformer | 16 | **98.61** |
| CIFAR10-DVS | PLIF (Fang et al., 2021) | 4Conv,2FC | 10 | 69.15* |
| | SLTT (Meng et al., 2023) | VGG-11 | 10 | 76.40* |
| | BA-LIF (Zhang et al., 2025a) | VGG-11 | 10 | 77.70† |
| | TRR (Zuo et al., 2024) | Spikingformer | 5 | 75.55 |
| | Spikformer(Zhou et al., 2022) | Spikingformer | 16 | 79.40* |
| | EventRPG(Sun et al., 2024) | EventRPG+VGG-11 | 10 | 84.90* |
| | **AND-LIF (ours)** | 4Conv,2FC | 10 | 69.79 |
| | | VGG-11 | 10 | 77.20 |
| | | Spikingformer | 4 | 77.70 |
| | | Spikingformer | 16 | 80.41 |
| | | EventRPG+VGG-11 | 10 | **85.60** |

that directly models the neuronal timescale as a time-varying function of a baseline value and neuronal signals:

$$\tau(t) = F\big(\tau_0, I(t), S(t)\big), \tag{11}$$

Here, $\tau_0$ serves as an intrinsic membrane time constant in the proposed formulation. Inspired by biological neurons that exhibit both long-term plasticity (Rimmer & Harper, 2006; Hong et al., 2016) and short-term dynamic modulation of their membrane neuronal timescales (Kuhn et al., 2004; Ha & Cheong, 2017), $\tau_0$ is treated as a learnable parameter

and optimized through backpropagation using surrogate gradients (Neftci et al., 2019). Based on this formulation, the function $F(\cdot)$ defines a unified modeling paradigm for adaptive temporal integration in spiking neurons.

Biological evidence suggests that strong synaptic drive and elevated firing activity shorten the membrane integration timescale, whereas weaker input and sparse firing lead to slower integration (Koch et al., 1996; Kuhn et al., 2004; Ha & Cheong, 2017; Gerstner & Kistler, 2002). Motivated by

these findings, we instantiate $F(\cdot)$ using two complementary activity-dependent feedback mechanisms. Specifically, we implement input-driven modulation and spike-driven modulation as additive reductions of the baseline time constant with bounded clipping, yielding stable and interpretable temporal dynamics.

$$\tau(t) = \text{clip}\big(\tau_0 - \tau_{\text{input}}(t) - \tau_{\text{output}}(t), \tau_{\text{min}}, \tau_{\text{max}}\big), \quad (12)$$

where $\tau_{\text{input}}(t)$ and $\tau_{\text{output}}(t)$ represent the adjustments induced by input intensity and output spiking activity, respectively. To capture both the persistence of past modulation and the influence of current signals, we define the input- and output-driven components as first-order recursive dynamics:

$$\tau_{\text{input}}(t) = \lambda_i \tau_{\text{input}}(t-1) + \eta_i I(t), \quad (13)$$

$$\tau_{\text{output}}(t) = \lambda_o \tau_{\text{output}}(t-1) + \eta_o S(t-1), \quad (14)$$

where $\lambda_i$ and $\lambda_o$ control the temporal persistence of past adjustments, while $\eta_i$ and $\eta_o$ determine the influence of current input and recent spiking activity, respectively. As illustrated in Fig. 1(b) and (c), these components jointly enable dynamic modulation of the membrane time constant.

Based on the dynamic time constant $\tau(t)$, the membrane decay factor and membrane potential update are given by

$$\rho_{\text{m}}(t) = e^{-\frac{\Delta t}{\tau(t)}}, \quad (15)$$

$$U^l(t) = \rho_{\text{m}}(t)\big(H^l(t-1) - S^l(t-1)V_{\text{th}}\big). \quad (16)$$

Together, these equations define an efficient spiking neuron model with adaptive neuronal dynamics. By modulating the neuronal timescale through a learnable baseline and activity-dependent terms, the proposed neuron can flexibly adjust its integration behavior across multiple timescales, achieving both short-term adaptability and long-term stability while maintaining low computational overhead.

## 4. Experiments

In this section, we evaluate the proposed AND-LIF neuron across a diverse range of dynamic benchmarks, to validate its effectiveness and generalization capability. To reduce randomness, we report the average results of three independent experiments in our experiments. For detailed information regarding the experimental setup, refer to Appendix A.1.

### 4.1. Results on Temporal Datasets

**DEAP Dataset.** In Tab. 1, we present a comparison of our method against existing SNN-based direct training approaches on the DEAP dataset. We follow the pre-possessing method used by (Zheng et al., 2024). The first 3s data of each trial is used for producing the average 1s baseline signal by averaging the 3s data per second. The following 60s data was normalized by subtracting the average 1s baseline signal every second and then divided into 20 segments of 3s for each. The table reports classification accuracies for both the Valence and Arousal dimensions, which correspond to the emotional positivity/negativity and intensity, respectively. As shown in Tab. 1, our model with an architecture of two fully-connected layers achieves SOTA accuracy, particularly on the challenging 3-class tasks, reaching 90.11% for Valence and 92.57% for Arousal, representing remarkable improvements of +9.08% and +9.97% over the baseline method (Sun et al., 2025). These results demonstrate the effectiveness of our approach in capturing complex affective patterns compared to prior SNN models.

**Neuromorphic Dataset.** We evaluate our AND-LIF neurons on three commonly used and effective SNN architectures: feedforward SCNN (Wu et al., 2018; Fang et al., 2021), VGG (Meng et al., 2023; Zhang et al., 2025a) and Spikingformer (Zhou et al., 2022). These architectures provide representative benchmarks for assessing the impact of our dynamic and learnable mechanism. As shown in Tab. 2, on both the DVS-Gesture and CIFAR10-DVS datasets, our AND-LIF consistently improves classification accuracy across various network configurations: it achieves an improvement of +9.68% over STBP (Wu et al., 2018) with a feedforward SCNN architecture at $T = 10$, and +3.12% over SLTT (Meng et al., 2023) with VGG-11 at $T = 5$ on DVS-Gesture, while yielding a gain of +2.25% over the Spikingformer baseline reported in (Zhou et al., 2022) on CIFAR10-DVS. These consistent enhancements demonstrate the generality and effectiveness of AND-LIF across different network designs, highlighting its applicability as a drop-in replacement for standard LIF neurons in a wide range of network scales.

### 4.2. Evaluation of Robustness Against Noise

To evaluate the robustness of our method against input perturbations, we introduce two types of noise widely adopted in robustness assessments of spiking neural networks (Li et al., 2020; Wu et al., 2022; Ding et al., 2024a): Gaussian noise and salt-and-pepper noise. Gaussian noise simulates continuous analog disturbances, while salt-and-pepper noise randomly corrupts a subset of pixels by setting them to either minimum or maximum values. Based on this motivation, we conduct robustness experiments on the DVS-Gesture dataset by imposing these noise types at various levels into the event streams, thereby examining the model's stability under both smooth and extreme corruptions.

Specifically, we subject the input events to Gaussian noise and salt-and-pepper noise, and compare the performance across four neuronal variants detailed in Appendix A.1.4: standard LIF, LIF with learnable $\tau$, LIF with dynamic $\tau$, and our proposed AND-LIF. The accuracy degradation curves

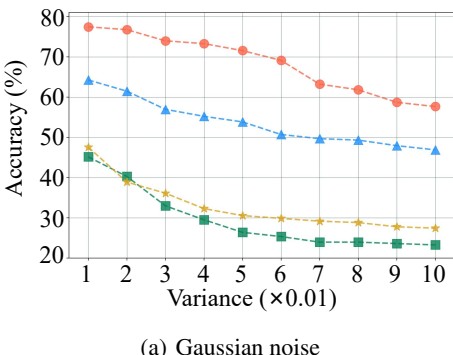
(a) Gaussian noise

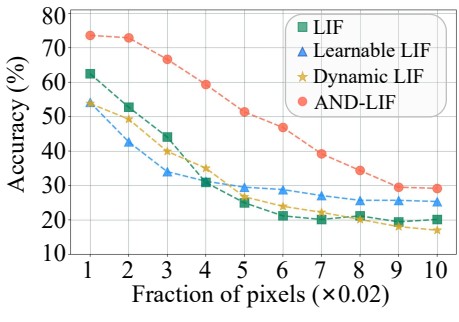
(b) Salt-and-pepper noise

*Figure 2.* Performance on DVS-Gesture under varying levels of (a) Gaussian noise, where the noise level reflects the variance of the zero-mean noise added to each pixel, and (b) salt-and-pepper noise, where the level indicates the fraction of pixels randomly replaced by either minimum or maximum intensity values. Detailed settings can be found in Appendix A.1.3.

*Table 3.* Comparison of energy consumption per timestep on the DEAP dataset. Arrows indicate the relative percentage increase (↑) or decrease (↓) compared to the standard LIF neuron. Accuracy improvements are reported as absolute differences (percentage points).

| Model | Firing rate (%) | Power (mJ) | Accuracy (%) |
| --- | --- | --- | --- |
| LIF | 46.68 | 0.441 | 79.06 |
| LTC-LIF (Yin et al., 2023) | 25.42 (↓45.5%) | 0.852 (↑93.2%) | 85.86 (+6.80) |
| BA-LIF (Zhang et al., 2025a) | 48.56 (↑4.0%) | 2.801 (↑535.2%) | 86.88 (+7.82) |
| **AND-LIF (ours)** | **12.79 (↓72.6%)** | **0.122 (↓72.3%)** | **90.11 (+11.05)** |

in Fig. 2 illustrate that AND-LIF maintains superior performance compared to the other variants, indicating that the combination of dynamic and learnable mechanisms effectively enhances robustness to input noise.

### 4.3. Evaluation of Energy Consumption

The quest for low power drives the development of spiking neural networks, making energy efficiency a central metric in neuromorphic computing. Following (Li et al., 2021b; Rathi & Roy, 2021; Zhang et al., 2025a), we calculate the firing rates and energy cost. To eliminate the influence of complex network architectures on energy estimation, we conduct the energy consumption analysis on the DEAP dataset using a consistent two-layer fully connected network as an example. This setup ensures that the results primarily reflect the contribution of the dynamic $\tau$ mechanism, rather than being biased by architectural complexity.

Since the introduction of dynamic $\tau$ mechanisms inevitably introduces additional computational overhead, we specifically compare among models that incorporate dynamic ntime constants. Therefore, we analyze the theoretical energy consumption of different neuron models, focusing on LIF (Sun et al., 2025), LTC-LIF (Hasani et al., 2021), BA-LIF (Zhang et al., 2025a), and our AND-LIF. Specifically, the energy is measured in 45nm CMOS technology where the multiply operation costs 3.7pJ energy and the accumu-

lation costs 0.9pJ energy (Horowitz, 2014). As shown in Tab. 3, our method significantly reduces computational operations compared to other dynamic approaches. Notably, although AND-LIF introduces additional computational operations, it substantially reduces the firing rate, resulting in an overall energy consumption of only 0.122 mJ while achieving 90.11% accuracy—a level that approaches one-quarter of the energy cost of the standard LIF model. This highlights the superiority of our approach in balancing computational cost and spiking efficiency.

### 4.4. Ablation Studies

**Influence of Dynamic Mechanism to Adaptability.** To further evaluate the contribution of the dynamic membrane time constant, we visualize its temporal evolution during inference on the DEAP dataset in Fig. 3(a). The figure shows that $\tau$ fluctuates within a certain range over time, reflecting input- and output-driven modulations. To further assess the importance of this dynamic behavior, we conduct an experiment where $\tau$ is fixed within this observed average range after training in Fig. 3(b). We find that disabling the dynamic updates results in an accuracy drop of over 10%, reducing performance to a level comparable to a standard LIF neuron. This suggests that the observed performance gains mainly arise from the temporal adaptation of $\tau$, rather than other factors.

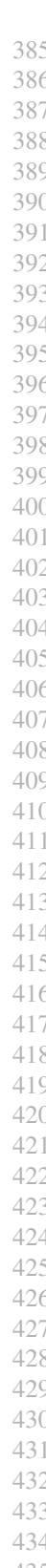
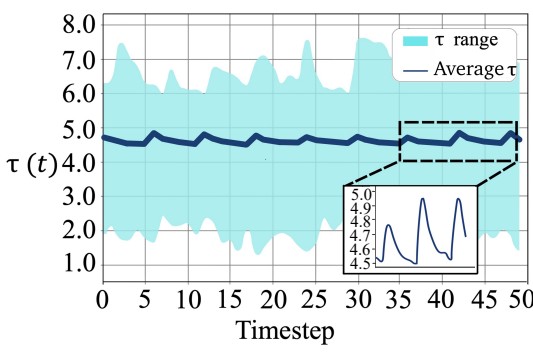
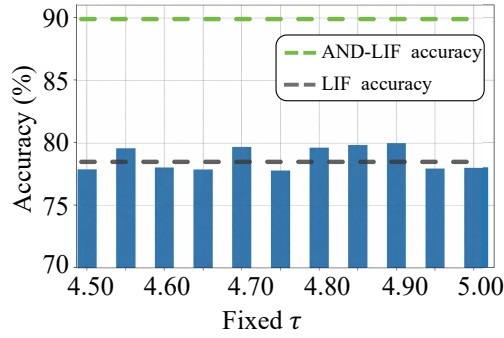

(a) Dynamic evolution

(b) Accuracy with fixed $\tau$

*Figure 3.* Ablation study on the dynamic $\tau$ mechanism. (a) Temporal evolution of $\tau$ in the AND-LIF neuron during inference. The shaded area indicates the fluctuation range caused by input- and output-driven modulation, with the solid line indicating the mean value. (b) Accuracy comparison with $\tau$ fixed within the fluctuation range, highlighting a significant performance drop when dynamic $\tau$ is disabled.

*Table 4.* Ablation study of learnable and dynamic $\tau$ in our AND-LIF. Values in parentheses indicate accuracy improvement (in percentage points) over the LIF neuron model.

| Dataset | Model | Accuracy (%) |
|---|---|---|
| DEAP | LIF | 79.06 |
| | + learnable $\tau$ | 82.18 (+3.12) |
| | + dynamic $\tau$ | 84.14 (+5.08) |
| | **AND-LIF** | **90.11 (+11.05)** |
| DVS-Gesture | LIF | 87.50 |
| | + learnable $\tau$ | 95.14 (+7.64) |
| | + dynamic $\tau$ | 95.49 (+7.99) |
| | **AND-LIF** | **96.18 (+8.68)** |

*Table 5.* Ablation study of input- and output-driven components and their combined effects. Values in parentheses indicate relative changes in firing rates and absolute accuracy improvements compared to the baseline model (Original Learnable $\tau_0$).

| Model | Firing rates (%) | Accuracy (%) |
|---|---|---|
| Original Learnable $\tau_0$ | 34.28 | 82.18 |
| $+\tau_{\text{output}}$ | 33.61 ($\downarrow$2.0%) | 83.30 (+1.12) |
| $+\tau_{\text{input}}$ | 13.41 ($\downarrow$60.9%) | 86.76 (+4.58) |
| $+\tau_{\text{input}}$ & $+\tau_{\text{output}}$ | **12.79 ($\downarrow$62.7%)** | **90.11 (+7.93)** |

**Influence of Dynamic and Learnable Mechanism.** To disentangle the contributions of the dynamic and learnable components in our AND-LIF neuron, we conduct ablation experiments on both the DEAP and DVS-Gesture datasets. The results in Tab. 4 show that both learnability and dynamics contribute positively to performance. Introducing either mechanism leads to a clear improvement over the baseline LIF. Furthermore, combining both leads to a further enhancement. In particular, AND-LIF achieves an accuracy gain of 11.05% on DEAP and 8.68% on DVS-Gesture over the standard LIF, validating our design choice of integrating dynamic flexibility with learnable adaptability into a unified neuron model.

**Influence of Input and Output Components.** To analyze the relative roles of input-driven and output-driven modulation in updating the membrane time constant $\tau$, we perform ablation experiments in which each component is selectively enabled or disabled on the DEAP dataset. As summarized in Tab. 5, introducing either component alone leads to consistent improvements in classification accuracy accompanied by reductions in firing rates, indicating that both mechanisms contribute to more efficient temporal integration. These results suggest that jointly considering input signals and spiking activity provides a more effective way to adapt the membrane neuronal timescale.

## 5. Conclusion

Inspired by intrinsic regulation mechanisms in biological neurons, we propose AND-LIF, a new neuron model that explicitly unifies a learnable base of neuronal timescale with intrinsically modulated dynamic regulation, enabling adaptive temporal dynamics that balance long-term stability and short-term adaptability within a computationally efficient formulation. Extensive evaluations across physiological and neuromorphic vision benchmarks demonstrate that such adaptive neuronal dynamics consistently improve accuracy, robustness, and efficiency across diverse SNN architectures. Together, these findings highlight adaptive neuronal dynamics as a viable and effective design pathway for advancing the performance and efficiency of neuromorphic systems.

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

# A. Appendix

## A.1. Experimental Details

### A.1.1. DATASETS

In this paper, we perform experiments on the EEG dataset DEAP and the neuromorphic datasets CIFAR10-DVS, DVS-Gesture.

DEAP (Koelstra et al., 2011): A multi-modal dataset for the analysis of human affective states. It contains electroencephalography and other physiological recordings from 32 participants, each watching 40 one-minute-long music videos. The dataset is labeled with ratings for arousal, valence, dominance, and liking, making it a standard benchmark for emotion recognition tasks using physiological signals.

DVS-Gesture (Amir et al., 2017): A neuromorphic benchmark dataset for gesture recognition, recorded using a dynamic vision sensor. It contains 1,464 instances of 11 different hand and arm gestures performed by 29 subjects under three different lighting conditions. The event-based nature of the data makes it ideal for testing the temporal dynamics and efficiency of spiking neural networks.

CIFAR10-DVS (Li et al., 2017): A neuromorphic version of the popular CIFAR-10 dataset, converted by displaying the original images on a monitor and recording the output with a dynamic vision sensor. It contains 10,000 event streams spanning 10 object classes, presenting a challenging benchmark for object recognition with event-based cameras that requires handling sparse, asynchronous inputs.

### A.1.2. TRAINING SETUP

Experiments are implemented using PyTorch (version 2.0.1) with CUDA 11.8 and conducted on a server equipped with NVIDIA RTX 4090 GPUs. In the experiment section, the details of the simulation time steps, training epochs, batch size, and some neuronal hyperparameters are summarized in Tab. 6. For the DEAP dataset, we adopt a two-layer fully connected

*Table 6.* Parameter settings on different learning tasks.

| Parameters | Descriptions | DEAP | DVS-Gesture | CIFAR10-DVS |
|---|---|---|---|---|
| Batch size | - | 200 | - | - |
| $T$ | Time steps | 6 | - | - |
| $\lambda_i, \lambda_o$ | Decay factor | 0.1 | 0.35 | 0.2 |
| $\eta_i, \eta_0$ | Activity impact factor | 0.01 | 0.1 | 0.05 |
| $V_{\text{th}}$ | Threshold | 1.0 | - | - |
| $N$ | Training epochs | 100 | - | - |

network structure with dimensions [512, 3/2], and the detailed hyperparameter settings are provided in Tab. 6. For the neuromorphic vision datasets DVS-Gesture and CIFAR10-DVS, as experiments were conducted across different network architectures, parameters such as the number of time steps and the firing threshold are not fixed, but instead follow the configurations used in previous works (Wu et al., 2018; Fang et al., 2021; Meng et al., 2023; Zhou et al., 2022). The only modification in these experiments is the replacement of the original LIF or PLIF neurons in the reference architectures with our proposed AND-LIF neurons for a fair comparison.

### A.1.3. ROBUSTNESS EXPERIMENTS

We evaluate the robustness of our model under input perturbations. This is critical for neuromorphic systems that operate in noisy environments, such as low-light settings or high-speed motion scenarios (Hendy & Merkel, 2022). We apply two widely used noise types: Gaussian noise and salt-and-pepper noise, both commonly used in robustness testing of SNNs (Li et al., 2020; Wu et al., 2022; Ding et al., 2024a).

**Gaussian noise:** We perturb the input by adding zero-mean Gaussian noise sampled from $\mathcal{N}(0, \sigma^2)$ to each pixel of the image. The standard deviation is controlled via the variance parameter $\sigma^2$. We define ten increasing noise levels as follows:

$$\sigma^2 \in \{0.01, \ 0.02, \ 0.03, \ 0.04, \ 0.05, \ 0.06, \ 0.07, \ 0.08, \ 0.09, \ 0.10\}.$$

Each noise level is applied independently to the test set, and model performance is reported as classification accuracy. This simulates gradually deteriorating sensor quality or transmission interference.

**Salt-and-pepper noise:** This noise simulates dead pixels and random corruption by flipping a proportion of image pixels to 0 (black) or 1 (white). For each level, we randomly select a fraction $a$ of pixels in the image and assign them new values drawn from a binary distribution. The tested levels are:

$$a \in \{0.02,\ 0.04,\ 0.06,\ 0.08,\ 0.10,\ 0.12,\ 0.14,\ 0.16,\ 0.18,\ 0.20\}.$$

This experiment tests the model's ability to recognize structure in partially corrupted input and maintain stability under non-Gaussian perturbations.

### A.1.4. NEURONAL MODEL VARIANTS

Specifically, we subject the input events to Gaussian noise and salt-and-pepper noise, and compare the performance across four neuronal variants: standard LIF, LIF with learnable $\tau$, LIF with dynamic $\tau$, and our proposed AND-LIF. The standard LIF neuron follows the formulation adopted in (Zheng et al., 2024).

As demonstrated by Eq. 4, the non-differentiable nature of spiking activity precludes the direct application of backpropagation to SNNs. To address this fundamental limitation, surrogate gradient methods replace the undefined gradients of spike generation with a differentiable surrogate function $u(\cdot)$ during error backpropagation. Specifically, while the spike activity is computed according to Eq. 4, its derivative with respect to the membrane potential is approximated as

$$\frac{\partial S^l(t)}{\partial U^l(t)} \approx u'\left(U^l(t), V_{\text{th}}\right), \tag{17}$$

which enables gradient-based optimization. We follow prior works (Wu et al., 2018; 2022; Ding et al., 2024b) and adopt a rectangular surrogate gradient function defined as

$$u'\left(U^l(t), V_{\text{th}}\right) = \frac{1}{a}\ \text{sign}\left(\left|U^l(t) - V_{\text{th}}\right| < \frac{a}{2}\right), \tag{18}$$

where $a$ is a hyperparameter controlling the width of the rectangular window. Under this surrogate gradient framework, model parameters that influence the membrane potential dynamics can be optimized via standard backpropagation. In particular, for the LIF with learnable $\tau$ variant, the membrane time constant is treated as a trainable parameter and updated jointly with synaptic weights through the same surrogate gradient mechanism during training, while remaining fixed during inference.

For the LIF with dynamic $\tau$ variant, we adopt the same dynamic time-constant formulation as described in Section 3.3. Specifically, the membrane timescale is modulated by both input intensity and recent spiking activity through input-driven and spike-driven components, while the baseline time constant $\tau_0$ is fixed and non-learnable in this variant. All other settings follow the formulation introduced in the main text.

## A.2. Detailed Description of the AND-LIF Neuron

The AND-LIF neuron extends the standard LIF model by introducing an adaptive membrane time constant that is modulated intrinsically by neuronal signals. At each time step, two auxiliary state variables are maintained to capture input-driven and spike-driven modulation, respectively. These variables are updated using first-order recursions and combined with a learnable base membrane time constant to compute the adaptive time constant, which is clipped to a predefined range to ensure numerical stability. The membrane decay factor is then recomputed at each time step based on the adaptive time constant, while the membrane potential update and spike generation follow the standard LIF formulation. Importantly, the AND-LIF neuron introduces no additional network branches and only a constant number of extra state variables, preserving computational efficiency. The complete update procedure of the AND-LIF neuron is summarized in Algorithm 1.

## A.3. Ablation Study on $\lambda$ and $\eta$ Coefficients

The selection of hyperparameters plays a critical role in overall network performance. To evaluate the impact of the key hyperparameters $\lambda$ and $\eta$ introduced in our AND-LIF model, we conduct a hyperparameter search on the DEAP dataset (three-class valence emotion classification task). We begin with a coarse-grained evaluation by setting both $\lambda$ and $\eta$ to values

---

**Algorithm 1** AND-LIF Neuron Update (Single Time Step)

---

**Require:** Input current $I(t)$, previous membrane potential $H(t-1)$, previous spike $S(t-1)$
**Require:** Previous modulation states $\tau_{\text{input}}(t-1)$, $\tau_{\text{output}}(t-1)$
**Require:** Learnable base time constant $\tau_0$
**Require:** Parameters $\lambda_i, \lambda_o, \eta_i, \eta_o$, bounds $\tau_{\min}, \tau_{\max}$
1: $\tau_{\text{input}}(t) \leftarrow \lambda_i \tau_{\text{input}}(t-1) + \eta_i I(t)$
2: $\tau_{\text{output}}(t) \leftarrow \lambda_o \tau_{\text{output}}(t-1) + \eta_o S(t-1)$
3: $\tau(t) \leftarrow \text{clip}\big(\tau_0 - \tau_{\text{input}}(t) - \tau_{\text{output}}(t), \tau_{\min}, \tau_{\max}\big)$
4: $\rho_m(t) \leftarrow \exp\left(-\frac{\Delta t}{\tau(t)}\right)$
5: $U(t) \leftarrow \rho_m(t)\big(H(t-1) - S(t-1)V_{\text{th}}\big)$
6: $H(t) \leftarrow U(t) + I(t)$
7: $S(t) \leftarrow \mathbb{I}\big(H(t) \geq V_{\text{th}}\big)$
8: **Return** $H(t), S(t), \tau_{\text{input}}(t), \tau_{\text{output}}(t)$

---

in $\{0.001, 0.01, 0.1, 0.2\}$. The corresponding accuracies—80.89%, 88.55%, 86.33%, and 79.88%—clearly indicate that very small (e.g., 0.001) or very large (e.g., 0.2) values significantly degrade performance, underscoring the importance of appropriate hyperparameter balancing. Based on these observations, we perform a finer-grained search within $\lambda$ in $\{0.01, 0.05, 0.1, 0.15\}$ and $\eta$ in $\{0.005, 0.01, 0.05, 0.1, 0.15\}$. As summarized in Tab. 7, the optimal performance is achieved at $\lambda$=0.1 and $\eta$=0.01, yielding 90.11% accuracy on the DEAP dataset. These results confirm that identifying suitable values for $\lambda$ and $\eta$ leads to substantially improved performance of the proposed neuron model.

*Table 7.* Accuracy under different settings.

| $\lambda$ \ $\eta$ | 0.005 | 0.01 | 0.05 | 0.1 | 0.15 |
|---|---|---|---|---|---|
| 0.01 | 88.71 | 88.55 | 86.72 | 87.65 | 86.52 |
| 0.05 | 87.81 | 88.98 | 88.55 | 87.34 | 87.77 |
| 0.1 | 88.77 | **90.11** | 87.46 | 86.33 | 86.41 |
| 0.15 | 87.34 | 88.86 | 87.42 | 85.62 | 83.71 |

