# OpenReview forum: "Intrinsically Adaptive Neuronal Dynamics Drive Accurate and Efficient Spiking Neural Networks"
_ICML.cc/2026/Conference — Submitted to ICML 2026_

### Official Review · Reviewer_kpJ9 · 2026-03-11

**Soundness:** 2
**Presentation:** 2
**Significance:** 3
**Originality:** 3
**Overall Recommendation:** 4
**Confidence:** 4

**Summary:**

The paper introduces a timescale adaptation mechanism in leaky integrate-and-fire (LIF) neurons (called AND-LIF). This timescale adaptation mechanism regulates the time constant of a LIF neuron based on the synaptic inputs it received and its own spiking activity. Using conductance-based modeling principles, the paper justifies the approach from a biological perspective. The AND-LIF model is then tested on 3 different benchmarks using several architectures and compared with state-of-the-art. The model is shown to outperform sota in all considered benchmarks. The paper then further compares the robustness against noise and energy consumption with sota and performs an ablation study to highlight the importance of the different mechanisms.

**Compliance With Llm Reviewing Policy:**

Affirmed.

**Final Justification:**

My two main concerns were about fairness in comparison with sota and biological representation. The authors have properly addressed my concerns, and the proposed additions will improve the quality of the manuscript.

**Key Questions For Authors:**

1. Why did you choose to only show the results for 3 classes BSNN (Sun et al., 2025) in Table 1, and not the results for BISNN nor the results for 2 classes BSNN? Are there other results from the referenced papers that were not disclosed in the Table?
2. Likewise, why was the performance of Spikformer with T=16 on DVS-Gesture not reported in Table 2?
3. Could you provide a systermatic comparison on the performance of AND-LIF, BA-LIF and LTC-LIF in the benchmarks of Tables 1 and 2?
4. In figure 2, the drop in performance of AND-LIF is similar than for other models, with a vertical shift due to higher initial accuracy. Does that not contradict the statement about enhanced robustness to noise?

**Limitations:**

Besides a hyperparameter sensitivity analysis, I did not find a discussion of the limitations in the manuscript.

**Strengths And Weaknesses:**

I find this paper well written. The research question, objectives, methodology and related work are clear. The proposed approach has merits, and experimental results show that the proposed approach outperforms the sota on 3 benchmarks. The different analyses permit to dig deeper into the inner workings of the proposed model. However, I have a few major concerns regarding the soundness of the work and its presentation.

My first major concern relates to the comparison with sota on the different benchmarks. Comparing the performance with sota is central to the proposed paper, and proposing a fair comparison is critical. In Table 1, the authors chose to present the results of Sun et al., 2025 for the BSNN model on 3 classes, which are outperformed by AND-LIF (81.03% vs 90.11% accuracy for valence, respectively). However, in the original paper, the BSNN model is proposed as an ablation study of the complete BISNN model, which reaches an accuracy of 90.72% on the same benchmark, similar to AND-LIF. This choice seems strange to me. Even if we consider that there is a good reason for this choice, then why are the results of BSNN for 2 classes not presented in Table 1? In the original paper, BSNN reaches 93+% for both valence and arousal, which is very similar to the reported accuracies for AND-LIF and outperforms the reported sota. In Table 2 - DVS-Gesture, the results of Spikformer with T=16 are not reported, although the authors reimplemented the model and use that configuration as the best AND-LIF model. Finally, the AND-LIF model is not compared to other LIF models with adaptive timescale (LTC-LIF and BA-LIF), except in Table 3 for one benchmark. I believe it is important to compare the performance with LTC-LIF and BA-LIF in a more systematic way.

My second major concern is about the presentation. I find the biological motivation overblown. The paper talks about "intrinsic regulation mechanisms in biological neurons" and uses a conductance-based modeling approach to show how synaptic inputs affect membrane time-constant. Although this is correct, biological membranes also contain many voltage-dependent ion channels that produce highly nonlinear changes in membrane conductance as the neuron is depolarized/repolarized. As stated in Koch et al., 1996: "(...) the classically defined tau(m) only provides a very rough estimate, typically an overestimate, of the response time of neurons and that alternative measures are required to capture the dependency of the time course of the membrane potential on ligand-gated and/or voltage-dependent membrane conductances." I am not pointing this out to criticize the approach taken in the paper, as I said earlier it has merit in the context of spiking neural networks, but as a call for caution in calling this mechanism biologically-inspired.

---

> ### Author Rebuttal · Authors · 2026-03-30
>
> We sincerely appreciate that the reviewer recognized the clarity of the paper, the effectiveness of the proposed approach, and the value of the analyses. We also value the important concerns on soundness and presentation, which can be addressed through clearer comparisons and more explicit discussion, as detailed below.
>
> ---
> **1. Fairness and completeness of benchmark comparisons (M1 & Q1 & Q2)**
> - Regarding BISNN, our main consideration was to **keep fair comparison under the same training paradigm**. Since BISNN follows an **ANN to SNN mapping approach**, whereas BSNN uses **direct training** (as ours), we considered BSNN to be the more appropriate in Table 1.
> - For the missing 2-class BSNN results, our original focus was mainly on the more challenging 3-class setting. To the best of our knowledge, BISNN and BSNN achieve very competitive performance on this task, and we therefore **prioritized** the 3-class comparison in the original table. For completeness of the task setting, we did include 2-class BSNN results in Table 1. When organizing the 2-class comparison, we mainly considered including methods different from those in the 3-class setting in order to **cover more related works**, and therefore overlooked the 2-class BSNN. Although the 2-class setting was not our main focus, if omitting these results could cause misunderstanding, we are very willing to include them to make the comparison more **complete and transparent**.
> - Likewise, for DVS Gesture, when preparing the comparison in Table 2, we mainly followed the settings reported in the original paper, which led us to overlook the Spikformer result with T=16.
> - We have now added these missing results to the revised table (Tabs. R1, R2 in https://anonymous.4open.science/r/AND-LIF-2405) to make the comparison **more complete**.
>
> ---
> **2. Biological motivation and presentation (M2)**
>
> We sincerely appreciate the reviewer’s recognition that the proposed approach has merit in the context of spiking neural networks, as well as the professional and careful caution regarding its biological interpretation.
> We fully understand the reviewer’s point that, although conductance based modeling provides a meaningful motivation for our design, real biological membranes also involve voltage dependent ion channels and **more complex nonlinear conductance effects**. We will incorporate this suggested important clarification into the revised manuscript. We believe this revision will make the presentation more complete and better grounded, and we sincerely thank the reviewer for this insightful reminder and for pointing us to this relevant perspective with **such a detailed description**.
>
> ---
> **3. Systematic comparison with adaptive timescale LIF variants (M1 & Q3)**
>
> A more systematic comparison with adaptive timescale LIF models is indeed necessary, and including these results makes both the soundness and the presentation of our work more complete. Following this valuable suggestion, we have added the corresponding comparison results to the revised tables (Tabs. R1, R2 in https://anonymous.4open.science/r/AND-LIF-2405),making the benchmarking of AND-LIF against other adaptive timescale LIF variants more systematic and transparent. The results are consistent with those in Table 3, and further demonstrate the advantages of AND-LIF in both **performance and efficiency**.
>
> ---
> **4. Clarification on robustness to noise (Q4)**
>
> We appreciate this important observation. Showing only the accuracies under noisy conditions may not be sufficient to fully support the robustness claim, since a higher clean accuracy can indeed lead to a vertical shift in the curves. To make this comparison more transparent, we additionally report the clean accuracy together with the noisy accuracies under Gaussian noise with variances 0.01 and 0.1 in the table below, representing medium and high intensity noise, respectively. We further compare the corresponding performance degradation relative to the clean setting. The results show that AND-LIF exhibits **smaller relative** performance degradation under noise while achieving higher absolute accuracies.
>
> |Model|Clean(%)|Medium(%)|High(%) |
> |-------------|-------------|----------------|-----------------|
> |LIF| 87.50|45.14 ($\downarrow$48.41%) |24.31 ($\downarrow$72.22%)|
> |Dynamic LIF|95.14|47.57 ($\downarrow$50.00%) |27.78 ($\downarrow$70.80%)|
> |Learnable LIF|95.49|64.24 ($\downarrow$32.73%) |45.49 ($\downarrow$52.36%)|
> |AND-LIF|96.18 |77.43 (**$\downarrow$19.49%**) |55.56 (**$\downarrow$42.23%**)|
>
> ---
> **5. Duscussing limitations**
>
> This concern, together with similar comments from other reviewers, has made us realize that the limitation discussion in the original manuscript is not sufficiently explicit. In the revised manuscript, we will add a dedicated discussion of the limitations of AND-LIF, including the current evaluation scope, the limitations of the present energy analysis, and the need for broader validation on more diverse tasks.

---

> > ### Author Rebuttal · Reviewer_kpJ9 · 2026-04-02
> >
> > I appreciate the author's thorough rebuttal. My concerns have been resolved, and I am adjusting my recommendation accordingly.

---

> > > ### Author Response · Authors · 2026-04-04
> > >
> > > We sincerely thank the reviewer for the positive update and for adjusting the recommendation after considering our rebuttal. We are very encouraged that our response was able to address your concerns.

---

### Official Review · Reviewer_a87E · 2026-03-12

**Soundness:** 1
**Presentation:** 2
**Significance:** 2
**Originality:** 3
**Overall Recommendation:** 4
**Confidence:** 4

**Summary:**

The paper  introduces an efficient LIF-based neuron model featuring a dynamically modulated membrane time constant.
The model is inspired by conductance-based synaptic interaction paradigms and their ability to "tune" the effective membrane time constant from a low- to a high-conductance regime, effectively modulating the leak rate of the membrane potential.
The presented scheme replaces the traditional conductance-based paradigm with a computationally much more efficient and trainable mechanism.
The authors benchmark their model on three datasets, all taken from the realm of "neuromorphic" challenges with an intrinsic temporal component.

**Compliance With Llm Reviewing Policy:**

Affirmed.

**Final Justification:**

I thank the authors for addressing my concerns through additions to the related works, additional experiments and a more extensive investigation of the energy consumption. I have raised my score.

**Key Questions For Authors:**

1. Could the authors discuss their model in the context of other (gated) RNNs and compare the performance with those?

2. Could the authors add the number of trainable parameters in the benchmarking tables (table 1 and 2) and ensure that the shown performance gains are not caused just by a larger number of trainable parameters?

3. I would encourage the authors to introduce additional datasets, especially ones with longer temporal context (e.g. DVS-Gestures seems to be solved with 5-10 timesteps, there is not much room for computationally powerful temporal processing). Commonly used datasets from the neuromorphic space would be SHD and SSC [3]

4. Could the authors clarify the estimate of energy efficiency, especially in light of the firing rate discrepancies? Was this a fair comparison?

**Limitations:**

yes

**Strengths And Weaknesses:**

**Presentation:**

The paper is generally well written and motivates the model by citing respective literature from computational neuroscience.
While the model is introduced well, this comes at the cost of introducing it only quite late in the paper (page 6).
The supporting figures are mostly sufficient.
Figure 1 is a bit confusing, as it seems to mix neuron models (panels a) and b)) with the specific modulation mechanism introduced by the manuscript (panel c)).
I would advice the authors to differentiate more clearly between the panels, potentially as a "zoom" into panel b).

**Significance:**

The significance of the presented work is hard to judge, mostly due to a lack of comparison with other models from the SSM and generally RNN literature (Question 1).
This is particularly relevant, as the modulation of the membrane time constants seems to resemble the gating from gated RNNs.
LTC networks [1] (already cited in the paper as being related) can serve as a good starting point for making the connection to more conventional ML models.
That said, the manuscript shows very convinving performance gains on the selected benchmarks, and I encourage the authors to extend their work on other benchmark datasets.

**Soundness:**

The authors perform an ablation study and are thus able to motivate their model quite well.
I am, however, missing
- a more detailed analysis of the temporal evolution of the time constants (not just the mean),
- a more thorough analysis of model complexity when comparing to other models on the benchmarking tasks (Question 2):
  In the current benchmarking tables (Table 1 and 2) the number of trainable parameters for the different models are not given.
  It is crucial here to not "just" match the architecture to the compared benchmarks but also ensure that you do not have an "unfair" advantage due to the added number of trainable parameters (i.e. ensure that the specific nature of the trainable parameters, not just their quantity, matters).
- a performance evaluation on a neuromorphic dataset which requires powerful, multi-timescale temporal processing (Question 3):
  While the DEAP dataset seems to operate on longer timescales (not clear how many timesteps are actually used per sample) the other datasets of DVS-Gestures and CIFAR-10-DVS only have a very short temporal structure (between 5 and 20 timesteps per sample).
  It is commonly accepted in the neuromorphic literature that DVS-Gestures can be solved with only minimal amounts of temporal processing [2].
  I think it would therefore be important to demonstrate the models capabilities on datasets that are known to require complex multi-timescale temporal processing.
  In the neuromorphic space the SHD and SSC datasets [3] are typically used for that.
- and a more transparent evaluation of the energy efficiency (Question 4):
  Here, the energy considerations seems to hinge significantly on the neuronal firing rates.
  Those however can be subject to regularization (often without significant performance penalties).
  Additionally, the addition of another dynamic variable (compared to the standard LIF) does not only incurr computational overhead, but also requires additional memory to hold that state-variable.
  It can cause significant energy costs to provide that memory space as well as continuously update it for every new timestep.
  This additional potential energy cost should be considered and compared for the different models considered.

**Originality:**

The authors draw inspiration from contemporary topics (neuron parameter heterogeneity, gating-like internal dymanics) in a novel and creative way, while founding it quite well in computational neuroscience literature.

[1] Hasani, R., Lechner, M., Amini, A., Rus, D., and Grosu, R. Liquid time-constant networks. In Proceedings of the AAAI Conference on Artificial Intelligence

[2] Perez-Nieves, Nicolas, et al. "Neural heterogeneity promotes robust learning." Nature communications 12.1 (2021): 5791.

[3] Cramer, Benjamin, et al. "The heidelberg spiking data sets for the systematic evaluation of spiking neural networks." IEEE Transactions on Neural Networks and Learning Systems 33.7 (2020): 2744-2757.

---

> ### Author Rebuttal · Authors · 2026-03-30
>
> Thanks for such a high recognition of the originality of our work, and your insightful comments.
> We understand that the reviewer has raised four key questions, which are further elaborated in the Significance and Soundness sections of the review. We respond to them below in order.
>
> ---
> **1. Discuss and compare with conventional ML RNN models (Q1)**
>
> We agree this important point that AND-LIF has a clear conceptual connection to gated RNNs and related models such as LTC networks, since they all regulate **the retention of temporal information** through mechanisms that depend on the **current input or internal activity**. Following this valuable suggestion to connect our work more explicitly to conventional ML RNN models, we further updated our related literature survey and further identified a recent and highly relevant work [1], which studies gated conductance-based neuron DGN and already provides a systematic comparison between **dynamic temporal mechanisms** in SNNs and conventional ML RNN models. Therefore, we will incorporate this connection more clearly in the revised manuscript rather than repeating an extensive discussion here.
> Furthermore, we conduct additional experiments on the SHD benchmark to compare our method with these related approaches. The corresponding results are shown in the table below, which further highlight the significant advantage of our method in terms of **low energy consumption while achieving high accuracy performance**.
>
> |Method|Accuracy (%)|Params (M)|Energy(nJ) |
> | ------- | ------------ | ---------- | ------ |
> |RNN|76.53|-|- |
> |LSTM|89.2|-|604.7|
> |DGN|87.78 |0.22|3.03|
> |AND-LIF|**89.58**|**0.17**|**1.52**|
>
> ---
> **2. Model capacity and fairness of comparison (Q2)**
>
> We would like to clarify the point about **parameter overhead**, as there may have been a **misunderstanding** regarding the model's complexity. In our method, the only additional trainable parameter introduced is the **layer-specific** base time constant $\tau_0$ (i.e., a **single one** parameter budget added per layer), so the overall **parameter overhead is almost negligible**. Following your suggestion, we have revised to add the number of trainable parameters in the result tables (see **Tabs. R1, R2** in https://anonymous.4open.science/r/AND-LIF-2405). These results show that the **performance gain is not caused by a larger number of trainable parameters**, but rather by the proposed **dynamic modulation mechanism** itself.
>
> ---
> **3. Scalability to longer temporal horizon benchmarks (Q3)**
>
> Based on your important suggestion on a stronger multi-timescale temporal processing performance evaluation, we additionally include experimental results on the SHD and SSC datasets. As shown in the table (as detailed in our response to **Reviewer j42t, point 3**), AND-LIF also achieves **better performance on these longer temporal benchmarks**, which further supports the effectiveness of our method beyond the relatively short horizon settings in the original manuscript.
>
> ---
> **4. Clarification on energy efficiency evaluation (Q4)**
>
> For energy evaluation, our estimate is not based on firing rates alone. Following prior works, we estimate energy from both the numbers of **multiplication and addition operations** and the neuronal **firing rates**, including the extra operations required to update the adaptive time constant in AND-LIF. We fully agree with the reviewer that extra dynamic state requires additional memory storage and updates at each time step, and that the associated costs, such as memory access, are not fully captured by this estimate. Since these costs are strongly **hardware dependent**, energy analysis in the field is often used as **a coarse reference** rather than an exact hardware level measurement. To provide a more complete picture, we therefore additionally report the training memory usage and the average training time per epoch (as detailed in our response to **Reviewer j42t, point 2**), so that the overhead introduced by the extra dynamic state can be understood more transparently.
>
> ---
> **5. Remaining minor concerns on presentation and exposition**
>
> We will improve the organization by introducing the proposed model earlier and making the overall presentation easier to follow. In addition, we note that the original Fig. 3 already shows not only the mean of $\tau$ but also its upper and lower bounds, namely the maximum and minimum values over time. Following your suggestion, we further analyze the contributions of the **input and output components** over time. At each time step, we sum $\tau_{\text{input}}$ and $\tau_{\text{output}}$ over one layer at each step and plot their normalized relative contributions over time. Fig. R1 in https://anonymous.4open.science/r/AND-LIF-2405 shows that the modulation of $\tau$ **adapts** to the specific input and spiking dynamics.
>
> ---
> [1] Bai, Q., et al. "A Brain-Inspired Gating Mechanism Unlocks Robust Computation in Spiking Neural Networks." ICLR (2026).

---

> > ### Author Rebuttal · Reviewer_a87E · 2026-04-03
> >
> > I thank the authors for addressing the points in my review. I have some remaining concerns:
> >
> > - I appreciate the additional discussion of RNNs of different types and the extension to the SHD dataset. Unfortunately, the proposed model does not seem to significantly outperform traditional gated RNNs (such as LSTMs). Also, the provided table is missing some information: what are the parameter numbers for the LSTM and RNN and how is the energy estimate for the LSTM calculated (why is it missing for the RNN)?
> > -  The authors have not fully addressed my concerns w.r.t. the energy estimates, especially regarding the internal dynamics and regularization to reduce overall firing activity. In particular, it would be important to compare how the relationship between firing rate regularization and resulting task performance looks like for the compared models. I agree that energy estimates are generally hard to get right – and should for that purpose not be used without experimental or simulation-based evidence.

---

> > > ### Author Response · Authors · 2026-04-04
> > >
> > > We are grateful for your insightful follow-up, which helps us better show the **dual** advantages of **accuracy and efficiency** that define our approach and are highlighted in our title.
> > >
> > > ---
> > > **1. Further Clarification on RNN Comparisons**
> > >
> > > We agree that, compared with traditional gated RNNs such as LSTMs, the performance gain of our method is **relatively modest** in terms of **accuracy**. However, we would like to further emphasize that one of the key advantages of our method is its **exceptional energy efficiency**, which we believe is an important contribution enabled by intrinsic neuronal adaptive dynamics.
> > >
> > > Regarding the missing information in the table, we apologize for the confusion. In the previous submission, we only included the results explicitly reported in the reference, which led to missing information for the LSTM/RNN baselines. Based on your suggestion, we have now supplemented the table with the corresponding data, including the parameter numbers for both LSTM and vanilla RNN.
> > >
> > > We also explicitly provide the energy estimation procedure for the LSTM in the updated table to avoid ambiguity. In the table, $m$ and $n$ denote the numbers of input and output neurons, respectively, and $F_{r_{\mathrm{in}}}$ and $F_{r_{\mathrm{out}}}$ denote the firing rates of the input and output neurons.
> > > |Model|Dynamics|Cost|Params (M)|Energy(nJ)|Accuracy (%)|
> > > |---|---|---|---:|---:|---:|
> > > |RNN|$h_t=\sigma_h(W_hx_t+U_hh_{t-1}+b_h)$|$n(m+n)E_{\text{MAC}}$|0.14|150.7|76.53|
> > > |LSTM|$f_t=\sigma_g(W_fx_t+U_fh_{t-1}+b_f)$; $i_t=\sigma_g(W_ix_t+U_ih_{t-1}+b_i)$; $o_t=\sigma_g(W_ox_t+U_oh_{t-1}+b_o)$; $\tilde c_t=\sigma_c(W_cx_t+U_ch_{t-1}+b_c)$; $c_t=f_t\odot c_{t-1}+i_t\odot\tilde c_t$; $h_t=o_t\odot\sigma_h(c_t)$|$(4mn+4nn+3n)E_{\text{MAC}}$|0.43|604.7|89.2|
> > > |DGN|$\rho_l^t=\varphi(1-g_1-C_l^{m,n}z_{l-1}^{t-1}-C_{l,\text{rec}}^{n,n}z_l^{t-1})$; $V_l^t=\rho_l^t\cdot V_l^{t-1}+W_l^{m,n}z_{l-1}^{t-1}+W_{l,\text{rec}}^{n,n}z_l^{t-1}-\vartheta z_l^{t-1}$|$(2mnF_{r_{in}}+2nnF_{r_{out}}+nF_{r_{out}})E_{AC}+nE_{\text{MAC}}$|0.22|3.03|87.78|
> > > |AND-LIF|Eqs.(12-14) in the manuscript|$(mnF_{r_{in}}+nF_{r_{out}})E_{AC}+(n+4mF_{r_{in}}+4nF_{r_{out}})E_{\text{MAC}}$|0.17|1.52|89.58|
> > >
> > > ---
> > > **2. Further Clarification on Energy Estimation and Firing-Rate Regularization**
> > >
> > > We fully agree with the reviewer that energy estimation in SNNs should be interpreted with caution. In our work, the reported energy values are intended as operation-level proxy estimates for relative comparison, rather than definitive measurements of hardware-level energy consumption. According to the reviewer’s comments, we have revised the manuscript to make these assumptions and limitations more explicit, and to avoid overstating the scope of the current energy analysis.
> > >
> > > Following the reviewer’s suggestion, we further performed a controlled regularization-based analysis using a standard and widely used firing-rate regularization strategy. Specifically, we applied the following target-rate regularization term to explicitly control the average firing activity:
> > > $$
> > > L=L_{\mathrm{task}}+\lambda_{\mathrm{rate}}(r-\rho)^2
> > > $$
> > >
> > > $$
> > > r=\frac{1}{TN}\sum_{t=1}^{T}\sum_{i=1}^{N}s_i(t)
> > > $$
> > > where $r$ denotes the average firing rate, $\rho$ is the target firing rate, and $s_i(t)\in\{0,1\}$ denotes the spike activity. Using this formulation, we regularized the LIF baseline so that it achieved a firing rate close to that of our AND-LIF, and then compared their task performance and energy proxy **under matched activity levels**.
> > >
> > > Consistent with the reviewer’s **insightful observation**, regularization can indeed reduce firing activity and thereby affect the resulting energy estimation. Notably, the results demonstrate that even when firing rates and energy costs are approximately matched, our AND-LIF maintains a **distinct accuracy advantage**.
> > >
> > > |Model|Firing Rate (%)|Energy (mJ)|Accuracy (%)|
> > > |---|---:|---:|---:|
> > > |LIF+Reg.|12.05|0.114|73.97|
> > > |AND-LIF (**internal dynamics only**)|12.79|0.122|**90.11**|
> > >
> > > Additionally, we tested an extreme case by setting the target firing rate to zero for both models. This allowed us to assess the **accuracy-energy trade-off** under the most **stringent sparsity levels**, further highlighting the advantages of the AND-LIF dynamics. The corresponding results are summarized in the table below.
> > >
> > > |Model|Firing Rate (%)|Energy (nJ)|Accuracy (%)|
> > > |---|---:|---:|---:|
> > > |LIF+Reg.|0.04|378.42|68.40|
> > > |AND-LIF+Reg.|0.038|362.47|**78.16**|
> > >
> > > These additional results clearly reflect that extreme sparsity does benefit energy efficiency, but accuracy is also inevitably affected. Notably, our model still maintained better performance.
> > >
> > > Given the limited scope of the rebuttal period, we present this as a **preliminary exploration**. A more systematic study of the interaction between internal dynamics and firing-rate regularization, following the reviewer’s technical insight, would be an interesting future topic worthy for further efforts.

---

### Official Review · Reviewer_j42t · 2026-03-15

**Soundness:** 3
**Presentation:** 2
**Significance:** 2
**Originality:** 2
**Overall Recommendation:** 4
**Confidence:** 4

**Summary:**

This paper proposes AND-LIF, a spiking neuron model with intrinsically adaptive temporal dynamics. The neuron model introduces a learnable timescale combined with both input-dependent and spike-dependent modulation, allowing neurons to dynamically adjust their temporal integration during inference. Experiments on EEG and neuromorphic vision datasets show that AND-LIF significantly improves accuracy, robustness to noise, and energy efficiency, achieving state-of-the-art results while reducing firing rates and power consumption. The work demonstrates that adaptive neuronal timescales are an effective design principle for improving spiking neural networks.

**Compliance With Llm Reviewing Policy:**

Affirmed.

**Key Questions For Authors:**

Most evaluations in this paper are conducted with relatively short temporal horizons (typically fewer than 20 time-steps). It would be interesting to understand whether the proposed neuron can scale to longer temporal sequences, such as speech tasks or sequence modeling tasks such as translation.

**Limitations:**

See in weaknesses

**Strengths And Weaknesses:**

Strengths

This paper introduces a novel spiking neuron model with adaptive neuronal time scales, enabling heterogeneous temporal dynamics within the network. The proposed AND-LIF neuron combines a learnable base time constant with activity-driven modulation, which enhances the expressiveness of neurons while encouraging sparse spiking activity.

The experimental evaluation covers multiple modalities, including EEG-based physiological signals and neuromorphic vision datasets. The proposed method consistently improves performance across several benchmarks and architectures, achieving competitive or state-of-the-art results compared with prior neuron models.

Weaknesses

There are several previous works that explore learnable or adaptive membrane time constants in spiking neurons. For example, PLIF [1] introduces learnable membrane time constants, and recent works such as [2] investigate heterogeneous temporal dynamics in SNNs. While the paper claims novelty in combining learnable base time constants with dynamic modulation, the distinction from existing approaches is not sufficiently clarified. It would strengthen the paper if the authors could more clearly explain that.

In the experiments, the proposed neuron is mainly compared with standard LIF neurons using the same architecture. However, AND-LIF introduces additional parameters and adaptive dynamics, which may increase the representational capacity of the model. For a fair comparison, it would be preferable to compare models with similar parameter counts or computational budgets.

It appears that the model is trained using BPTT. Since the proposed neuron introduces additional dynamic states and recursive updates, it may increase both memory consumption and computational cost during GPU training. Could the authors provide more details on the training overhead introduced by the proposed neuron dynamics, such as training time, memory usage, or scalability compared to standard LIF neurons?

[1] Fang, Wei, et al. "Incorporating learnable membrane time constant to enhance learning of spiking neural networks." *Proceedings of the IEEE/CVF international conference on computer vision*. 2021.

[2] Zheng, Hanle, et al. "Temporal dendritic heterogeneity incorporated with spiking neural networks for learning multi-timescale dynamics." *Nature Communications* 15.1 (2024): 277.

---

> ### Author Rebuttal · Authors · 2026-03-30
>
> Thanks for your positive assessment and constructive comments, and we hope the following responses can address your concerns.
>
> ---
> **1. Further clarifying distinction from existing approaches (W1)**
>
> Following your valuable suggestion, we have revised to further clarify our distinction from the prominent works, including PLIF [1] and DH-LIF [2] that both have made important contributions to the study of learnable membrane time constants in spiking neurons.
>
> These existing works significantly advanced the modeling of neuronal timescales in SNNs, but their time constants remain **learnable but fixed** during inference. Our method is built upon these important developments, and **further extends** them from learnable membrane time constants to **adaptive** membrane time constants. Specifically, AND-LIF combines a learnable base time constant with input-driven and spike-driven modulation, making its membrane time constant both **learnable and dynamically** adjusted at each time step during inference. We will make this distinction clearer in the revised manuscript.
>
> ---
> **2. Concerns on parameter overhead and computational budgets (W2 & W3)**
>
> We would like to clarify the point about parameter overhead (**W2**), as there may have been a **misunderstanding** regarding the model's complexity. In fact, the only additional learnable parameter introduced by AND-LIF is the **layer-specific** base time constant $\tau_0$ (i.e., a **single one** parameter budget added per layer), so the overall **parameter overhead is almost negligible**.
>
> Due to space constraints, we provide an abbreviated table below detailing the parameter counts (see column 'Params'). The complete table, containing the full set of results, can be found in the anonymous repository: https://anonymous.4open.science/r/AND-LIF-2405 (**Tab. R1 & R2**).
>
> For computational overhead (**W3**), following your advice, we added experiments on GPU memory usage and average training time per epoch to quantify the practical training cost introduced by AND-LIF. As shown in the table below, on the DEAP dataset, standard LIF requires 540 MB GPU memory and 2.30 s per epoch, while AND-LIF requires 550 MB and 2.86 s per epoch. These results show that, the additional dynamic states only introduce marginal overhead, and **the increase remains modest in practice**.
>
> These suggested additional results underscore the **computational efficiency** of our approach. The proposed adaptive dynamics are realized through **lightweight** recursive updates rather than complex structural additions, yielding **substantial accuracy improvements** with minimal overhead.
>
> |Model|GPU Memory (MB)| Avg. Time / Epoch (s)|Params| Accuracy (%) |
> |-------| --------------- |---------------------| ------------ | ------------ |
> |LIF|540|2.30|1050627| 79.06|
> |AND-LIF|550 (**+10**)|2.86 (**+0.56**)|1050629 (**+2**)|90.11(**+11.05**)|
>
> ---
> **3. Scalability to longer temporal sequences (Q1)**
>
> Based on this important suggestion, we additionally evaluate the proposed neuron on the SHD and SSC speech tasks. Specifically, these experiments are conducted based on the open source implementation provided in [3], into which we incorporate our AND-LIF neuron. We also understand the reviewer’s concern regarding **parameter overhead** and **computational budgets**, and therefore report the corresponding parameter, memory, and training time results together with these additional experiments. As shown in the table below, AND-LIF **achieves consistent performance on these longer temporal horizon benchmarks**, which further supports the effectiveness of our method beyond the relatively short horizon settings in the original manuscript.
>
> |Dataset|T|Model|Params(M)|Memory (MB)|Time / Epoch (s)|Accuracy (%)|
> | ------- | --- | ------- | ---------- | ----------- | ---------------- | ------------- |
> | SHD|100|LIF|0.17|2368|18.99|87.88 |
> |||AND-LIF|0.17|2496 (**+128**)|20.59 (**+1.60**)|89.46 (**+1.58**)|
> |SSC|100|LIF|0.98|3304|110.29|77.83|
> |||AND-LIF|0.98|3406 (**+102**)|114.64 (**+4.35**)|79.81 (**+1.98**)|
>
> ---
> [1] Fang, Wei, et al. "Incorporating learnable membrane time constant to enhance learning of spiking neural networks." ICCV (2021).
>
> [2] Zheng, Hanle, et al. "Temporal dendritic heterogeneity incorporated with spiking neural networks for learning multi-timescale dynamics." Nature Communications (2024).
>
> [3] Wang, J., et al. "Efficient Speech Command Recognition Leveraging Spiking Neural Networks and Progressive Time-scaled Curriculum Distillation." Neural Networks (2025).

---

> > ### Author Rebuttal · Reviewer_j42t · 2026-04-05
> >
> > The rebuttal has clarified my main concern. I appreciate the authors’ efforts on additional evaluations and explanations.

---

> > > ### Author Response · Authors · 2026-04-07
> > >
> > > We sincerely thank the reviewer for the positive feedback. We are pleased that our rebuttal helped clarify the main concern, and we appreciate the reviewer’s acknowledgment of our additional evaluations and explanations.

---

### Official Review · Reviewer_9BME · 2026-03-22

**Soundness:** 3
**Presentation:** 2
**Significance:** 3
**Originality:** 3
**Overall Recommendation:** 4
**Confidence:** 4

**Summary:**

This paper proposes a new LIF neuron model that is inspired from the biological neurons' intrinsic regulation of the membrane timescale. Authors propose a dynamic timescale ,$\tau(t)$, which updates itself dynamically based on the input intensity and spike activity at each time step, making it dependent on both the input and output activity on a temporal level. The experimental results show that this dynamic modulation not only shows higher accuracy than the other baselines in different benchmarks, but also reduces energy consumption and firing rates.

**Compliance With Llm Reviewing Policy:**

Affirmed.

**Final Justification:**

I appreciate the authors' effort for answering the questions during the rebuttal. Even though they resolved the questions I had, considering the paper's clarity, organization and significance of the experiments done, I do not think that the paper is at the acceptance level, since I do not see adequate performance gain or comparison/ablation study with the conventional recurrent models & more complex temporal datasets.  The proposed model does not seem to outperform significantly with the presented additional comparisons during the rebuttal. I will keep my score (4: weak accept)

**Key Questions For Authors:**

1. What is the initial value of learnable $\tau_0$, how is it initialized? Also the same question for the $\tau_{input}$ and $\tau_{output}$ when $t=0$, could you elaborate more? They are not introduced well in the paper.

2. Does integration of input and output activity affect equally when it comes to modulating the dynamic time constant? How do they impact the time constant? It would be good to show the history of $\tau_{input}$ and $\tau_{output}$ independently from each other.

3. Time constant bounds are not explained clearly, how are they initialized and based on what?

4. Does increasing or decreasing the time steps $T$ have any impact on how the time constant $\tau$ is dynamically changed through time? Could you elaborate?

**Limitations:**

Authors should discuss the limitations of the AND-LIF approach clearly, especially when it comes to more complex architectures and benchmarks. It should be also discussed whether this approach can be used for tasks other than the classification.

**Strengths And Weaknesses:**

Strengths:

++ The approach is clear, understandable and applicable to other benchmarks.

++ A simple yet effective formulation for LIF neurons, keeping the computational overhead almost the same as the others.

++ Energy efficient neuron model.

Weaknesses:

--  Presentation of the experimental section is not clear and comprehensive. Before diving into results, main baselines should be explained at least in couple of sentences, what are the differences between your approach and the baselines, since not all the baselines are discussed in the related work.

-- Methods section is unbalanced and hard to read, with unclear parts, could have been shorter and more focused on the authors approach rather than the simple LIF and conductance-based neurons.

-- Ablation study could have been broader, especially the importance of dynamic $\tau$ and the how the input and output time constants impact it.

Other comments:
- In Section 4.1 Neuromorphic Dataset, you stated that AND-LIF improves upon STBP +9.68% where T=10, but from the Table 2., the difference seems to be +8.68% (AND-LIF=96.18 vs. STBP=87.50). Probably it is a typo.
- Typo in Line 378 (ntime)

---

> ### Author Rebuttal · Authors · 2026-03-30
>
> We thank the reviewer’s positive assessment of our work's originality and significance. Furthermore, we are grateful for the constructive feedback aimed at enhancing the paper's technical depth and clarity. Our responses are organized below, starting with the most substantial points raised.
>
> ---
> **1. Clarification on setup details (Q1 & Q3)**
>
> Given the reviewer’s emphasis on the experimental setup, providing further clarification on these details is our **highest priority**. Accordingly, we address this aspect first.
>
> （Re **Q1**） To ensure a fair comparison, we use the **same initial** $\tau$ as the baselines under the same architecture, so the performance difference comes from dynamic modulation rather than initialization. For $t=0$, both $\tau_{\text{input}}(t)$ and $\tau_{\text{output}}(t)$ are initialized to $0$ and are then computed according to Eq. (13) and Eq. (14). Specifically, the input related term at $t=0$ is $\tau_{\text{input}}(0)=\eta_i I(0)$. Since the neuron has not generated any output spike at this time, we have $\tau_{\text{output}}(0)=0$.
>
> （Re **Q3**） Considering the physical meaning of $\tau$ and the computation $\rho_{\mathrm{m}}(t)=e^{-\frac{\Delta t}{\tau(t)}}$, $\tau(t)$ should remain positive. Therefore, the lower bound of $10^{-3}$ is used to prevent invalid or unstable values caused by non positive $\tau$. For the upper bound, we do not impose an additional constraint, since when $\tau(t)>0$, the exponential term in $\rho_{\mathrm{m}}$ is naturally bounded in $(0,1)$.
>
> ---
> **2. Does integration of input and output terms affect equally? (Q2 & W3)**
>
> In short, our results indicate that they **do not**. Ablation results in Table 5 reflect that the two terms play distinct roles in performance by modulating $\tau$. Following your suggestion to better visualize their influence on $\tau$ over time, we have added a new analysis to quantify their relative contributions (see the **Fig.R1** in https://anonymous.4open.science/r/AND-LIF-2405
> ). Specifically, at each time step, we aggregate the contributions of $\tau_{\text{input}}$ and $\tau_{\text{output}}$ across all neurons within a layer, normalize them by their sum, and plot their relative contributions over time. The results show that **their contributions vary over time**, indicating that the dynamic time constant is **jointly modulated by both factors rather than equally determined by them**.
>
> When $T=6$, each input sample spans 6 time steps, so the input term contributes 100% at the first time step of each sample, namely time steps 1, 7, and 13 in Fig.R1. Different samples further exhibit different contribution patterns, showing that the modulation of $\tau$ **adapts** to the specific input and spiking dynamics.
>
> ---
> **3. Does changing time steps affect the evolution of $\tau$ ？(Q4)**
>
> In dynamic datasets, changing the number of time steps does not only change the simulation length, but also changes how the input sequence is temporally constructed. As a result, the neuronal dynamics of the whole network will also change. Since the adaptive modulation in AND-LIF is strongly dependent on neuronal activity, **changing the time steps will naturally affect the evolution of $\tau$**. To further clarify this point, we additionally conduct an experiment under $T=12$, and the corresponding results are shown in Fig. R2 (see Fig. R2 in https://anonymous.4open.science/r/AND-LIF-2405
> ). Compared with the result under $T=6$ (Fig. 3 in the manuscript), the temporal characteristics of the neuron do change. At the same time, it can also be clearly observed that within one $T$ period, the neuron still exhibits a **similar dynamic pattern**.
>
> ---
> **4. Discussing limitations**
>
> We agree that explicitly discussing the limitations and future scope of our approach will provide a more comprehensive view of the work. While AND-LIF is a general-purpose, neuron-level mechanism not restricted to any specific architecture, our initial validation has focused on representative classification tasks to clearly demonstrate its core dynamics. We acknowledge that its application to more complex architectures and tasks beyond classification represents a vital direction for future research. We have added this in the revised paper and appreciate your suggestions.
>
> ---
> **5. Remaining minor points (W1, W2 & other comments)**
>
> Thanks for your detailed comments for improving the quality and clarity of our manuscript. We have revised to better explain baseline selection, clarify the main differences from the compared baselines, shorten the LIF and COBA-LIF background, and correct the typo in Section 4.1 (+8.68% rather than +9.68%) and the typo “ntime” in Line 378.

---

> > ### Author Rebuttal · Reviewer_9BME · 2026-04-03
> >
> > I want to thank authors for detailed explanations to the questions I asked. I don't have any other concerns left. I will adjust my recommendation accordingly.

---

> > > ### Author Response · Authors · 2026-04-04
> > >
> > > We sincerely thank the reviewer for the positive feedback. We are deeply encouraged by the reviewer’s positive assessment and glad that our responses helped clarify the questions and address the concerns.

---

### Decision · Program_Chairs · 2026-04-30

**Decision:**

Reject

**Comment:**

This paper focuses on improving the expressivity of spiking neural networks (SNNs) by making neuronal dynamics adaptive over time, rather than keeping fixed membrane timescales. It introduces AND-LIF, which is essentially a modified LIF neuron with a learnable base timescale and additional input- and spike-dependent modulations, in order to allow dynamic adjustment of how neurons integrate temporal information during processing. The core idea is that neurons can flexibly adjust their behavior across multiple timescales depending on the relevant input context, rather than relying on a single fixed timescale.

The paper went through some discussions during the rebuttal phase. While the reviewers agreed that the proposed approach is simple and yet interesting and effective, they also raised serious concerns about the paper's empirical extent and presentation of key results, leading to borderline scores on the positive side with no strong confidence towards a solid acceptance decision after discussions.

One of the main concerns was regarding scalability to longer temporal sequences and additional experiments on that end, which the authors addressed via preliminary experiments on SHD and SSC where AND-LIF was compared against vanilla LIF models, although comparisons to SoTA methods on these widely-used benchmarks remains open. It is also unclear how AND-LIF based models truly harness dynamically regulated timescales under very short simulation durations (T<16) as used in the main experiments. The AC recommends that additional experiments with larger timesteps, such as the SHD and SSC ones presented during the rebuttal, be extensively explored in comparison with other SoTA methods.

Notably, another raised concern was regarding the selective presentation of benchmark results. The paper seemed to omit certain baseline results in settings where they are competitive (e.g., BSNN in the 2-class setting) while retaining less favorable comparisons, by overlooking relevant hybrid ANN-to-SNN approaches that could very well serve as reasonable benchmarks. The fact that the majority of these omissions only came to light during the rebuttal makes the contributions appear less reliable. The AC therefore requests further revisions that openly discuss how the proposed method differs from existing approaches, justify which SoTA comparisons are included or omitted, and provide transparency around each of the presentation choices made in the paper.

One further point of concern was the unclear empirical distinction between the proposed approach and existing SNN spike-frequency adaptation mechanisms, or simply using learnable membrane decay time constants, which are techniques already exploited in several recurrent SNN models. As one reviewer noted, the novelty over prior work on learnable or adaptive membrane time constants (e.g., PLIF, DH-LIF) is also not clearly established in the manuscript, and the paper lacks sufficient discussion distinguishing its contributions from existing methods, indicating that clarifying revisions are needed on this front.

Overall, the AC believes that the paper cannot be accepted to the conference at this time due to several open ambiguities and necessary major revisions. The identified concerns, particularly around empirical evaluation rigor and openness, should be addressed in a future revision with significant changes to the empirical content of the manuscript.